

# Multi-model effective radiative forcing of the 2020 sulphur cap for shipping.

Ragnhild Bieltvedt Skeie[1], Rachael Byrom[1], Øivind Hodnebrog[1], Caroline Jouan[1], Gunnar Myhre[1]

[1]CICERO Center for International Climate Research, Oslo, Norway

*Correspondence to*: Ragnhild Bieltvedt Skeie (r.b.skeie@cicero.oslo.no)

**Abstract.** New regulations of sulphur emissions from shipping were introduced in 2020, reducing emissions of $SO_2$ from international shipping by ~80%. As $SO_2$ is an aerosol precursor, this drop in emission over the ocean will weaken the total aerosol effective radiative forcing (ERF) that historically has masked an uncertain fraction of the warming due to increased concentration of greenhouse gases in the atmosphere. Here, we use four global climate models and a chemistry transport model to calculate the ERF due to an 80% reduction in $SO_2$ emissions from international shipping relative to 2019 emission estimates. The model means of the ERF range from 0.06 to 0.09 W m$^{-2}$ corresponding to the ERF due to the increase in $CO_2$ concentration over the last two to three years.

## 1 Introduction

On 1 January 2020 the International Maritime Organization (IMO) regulation on the limitation of sulphur in shipping fuel fuel (IMO, 2018) entered into force. This rule, hereafter named IMO2020, reduces the allowed sulphur content of marine fuels used outside of emission control areas from 3.5% to 0.5%. To fulfil the regulation ships must either use low sulphur fuel or install scrubbers that remove $SO_2$ from the exhaust. According to IMO, this new regulation will lead to a 77% reduction in total $SO_2$ emissions from shipping (IMO).

Emission of sulphur dioxide ($SO_2$) from shipping impacts air quality, is harmful for human health, has negative consequences for ecosystems and affects climate (Eyring et al., 2010). Through chemical reactions in the atmosphere, the emitted $SO_2$ is transformed into sulphate particles. Sulphate particles can alter the energy balance at the top of atmosphere (TOA), directly by reflecting solar radiation or indirectly by modifying cloud properties, which are quantified by the effective radiative forcing (ERF) of aerosol-radiation-interaction (ERFari) and aerosol-cloud-interaction (ERFaci), respectively.

$SO_2$ is the dominant contributor to the aerosol ERF and was assessed to be $-0.94 \pm 0.69$ W m$^{-2}$ (90% confidence interval) in 2019 relative to 1750 with a dominant contribution from ERFaci (Szopa et al., 2021). The total aerosol ERF is assessed to be $-1.3 \pm 0.7$ W m$^{-2}$ (90% confidence interval) (Forster et al., 2021).



The total sulphur emissions from international shipping in 2019 was 10.9 Tg $SO_2$ $yr^{-1}$ which is 13% of total anthropogenic emissions of $SO_2$ (O'Rourke et al., 2021). The $SO_2$ emissions from international shipping increased steadily from the 1980s to a maximum in 2008 of 14 Tg $yr^{-1}$ $SO_2$ (Fig. S1). Over the same period the total anthropogenic $SO_2$ emission decreased, and the relative contribution of international shipping emissions increased. Assuming an 80% reduction of $SO_2$ emissions in
2020 from the shipping sector, the emission would have dropped by 8.7 Tg $yr^{-1}$ corresponding to a 10% reduction in the total anthropogenic $SO_2$ emissions in 2019.

Eyring et al. (2010) summarized previous studies of the radiative forcing of the shipping sector and came up with a best estimate of the $SO_2$ radiative forcing for the direct effect (aerosol-radiation interaction) of -0.031W $m^{-2}$ in 2005 while the aerosol radiative forcing indirect effect (aerosol-cloud interaction) estimates ranged from -0.066 W $m^{-2}$ (Fuglestvedt et al.,
2008) up to -0.6 W $m^{-2}$ (Lauer et al., 2007). However, the estimate of Lauer et al. (2007) included other aerosols in addition to sulphate and had a large sensitivity to the assumption of geographical emission distribution. Subsequently, their estimate was as large as 39% of the total indirect aerosol radiative forcing (Eyring et al., 2010).

Several studies have looked at the radiative effects of sulphur emission reduction from international shipping that correspond to IMO2020. Note that this regulation was discussed and adopted by the IMO several years prior to its implementation,
allowing for the modelled projection of its associated climate impact. Partanen et al. (2013) found a large shipping radiative forcing of 0.33 W $m^{-2}$ following an 89% reduction in 2020 $SO_2$ emissions relative to 2010 using the ECHAM-HAMMOZ model. Jin et al. (2018) calculated a 0.23 W $m^{-2}$ forcing following a reduction in shipping $SO_2$ fuel content from 3.5% to 0.5% using CESM1.2.2 and Sofiev et al. (2018) reported a forcing of 0.071 W $m^{-2}$ using the SILAM chemical transport model that accounted for the direct aerosol effect and the effect of lowering the cloud albedo. More recently, Bilsback et al.
(2020) found a smaller ERF of just 0.027 W $m^{-2}$ for an 85 % reduction in $SO_2$ emissions from the shipping sector using GEOS-Chem-TOMAS.

The effect of ship emission regulations can also be observed from space via ship tracks. These provide a visualisation of aerosol-cloud interactions along the route of travel due to the enhanced concentration of sulphate particles which act as cloud condensation nuclei and cause an increase in the number of droplets within a cloud. A higher number of droplets makes the
cloud brighter and identifiable in satellite imagery. Yuan et al. (2022) and Watson-Parris et al. (2022) both reported a reduction in the presence of ship tracks after IMO2020 entered into force. However, cloud properties where ship tracks are not visible are also found to be impacted by ship emission (Manshausen et al., 2022), therefore studying ship tracks alone to estimate ERFaci will introduce selection biases (Glassmeier et al., 2021).

In this study, we use four different atmospheric chemistry climate models and one chemistry transport model to diagnose the
ERF of an 80% reduction in shipping $SO_2$ emissions corresponding to the world-wide emission reductions associated with IMO regulations that came into play in 2020.



## 2 Method

To calculate the ERF of the 2020 shipping emission cap, four global climate models (Table 1) is used to perform two atmosphere-only simulations: one baseline integration with 2019 anthropogenic aerosol (and precursor) emissions and one
perturbed integration where $SO_2$ and $SO_4$ emissions from shipping are reduced by 80%. The ERF is then calculated as the difference in top-of-atmosphere net radiative flux between the two simulations. As recommended by the Radiative Forcing Model Intercomparison Project (RFMIP, Forster et al., 2016;Smith et al., 2020), ERF is diagnosed using fixed sea-surface temperatures and sea ice climatology. We run two ensemble members for each global climate model for the simulation length specified in Table 1. This length needs to be sufficiently long to reduce the signal to noise ratio in the ERF calculation
(Forster et al., 2016), given that the potential impact of IMO2020 on radiative fluxes could be relatively small (e.g., Jin et al., 2018;Sofiev et al., 2018;Bilsback et al., 2020).

We further use a CTM (Table 1) and an offline radiative transfer model to calculate the RF of aerosol-cloud interactions and aerosol-radiation interactions associated with the 2020 shipping emission cap.

In all simulations we use 2019 anthropogenic aerosol (and precursor) emissions from the Community Emissions Data
System (CEDS) (v_2021_02_05, O'Rourke et al., 2021) which builds upon the CEDS inventory described in McDuffie et al. (2020), hereafter named CEDS_v2021.

**Table 1: Models included in the study, number of ensembles and length of the simulation. Note that OsloCTM3 is a CTM run with fixed meteorology and one year is sufficient for calculations of radiative forcing.**

| Model | #Ensemble members | Simulation length |
|---|---|---|
| CESM2 | 2 | 100 years |
| ModelE_MATRIX | 2 | 50 years |
| ModelE_OMA | 2 | 50 years |
| NorESM2 | 2 | 200 years |
| OsloCTM3 | 1 | 1 year |


In the following a short description of each model and their individual configuration and set up is provided.

## 2.1 CESM2

The Community Earth System Model 2 (CESM2) has been used with the Community Atmosphere Model Version 6 (CAM6)
atmospheric module (Danabasoglu et al., 2020). Simulations have been performed using fixed monthly sea-surface temperatures and sea-ice concentrations representing year 2000 climatology (ensemble member 1) and 2010 climatology (ensemble member 2). Greenhouse gas concentrations are representative of year 2000 climatology while anthropogenic



aerosol (precursor) emissions are from year 2019 from the CEDS_v2021. The model has been set up with a horizontal resolution of 1.9°x2.5° and 32 vertical layers. Aerosols physics are treated according to the Modal Aerosol Module version 4

(Liu et al., 2016) and aerosol-cloud interactions are included.

## 2.2 NASA GISS ModelE

Two versions of the Goddard Institute of Space Sciences Earth System Model (GISS ModelE version 2.1.2) developed by NASA (National Aeronautics and Space Administration) (Kelley et al., 2020) were used. The two versions, differing in the aerosol scheme, i.e., the two-moment MATRIX (Multi-configuration Aerosol Tracker of Mixing State) scheme in which all

aerosols are internally mixed (Bauer et al., 2020;Bauer et al., 2008) hereafter named ModelE_MATRIX and the One-Moment Aerosol (OMA) scheme (Koch et al., 2006) hereafter named ModelE_OMA, in which aerosols are assumed to remain externally mixed. MATRIX and OMA only include the first indirect effect (Nazarenko et al., 2017). As ModelE_OMA and ModelE_MATRIX have different aerosol physics, we treat these as two separate models in this study.

All simulations were conducted using fixed monthly sea-surface temperatures and sea-ice concentrations, reflecting the

climatology of the year 2000 (ensemble member 1) and 2010 (ensemble member 2). Greenhouse gas concentrations are representative of the year 1994 climatology, while anthropogenic aerosol and aerosol precursor emissions are the CEDS_v2021 emissions for year 2019. Both ModelE_OMA and ModelE_MATRIX have a horizontal resolution of 2.0° in latitude by 2.5° in longitude and 40 vertical layers, extending from the surface to 0.1 hPa.

## 2.3 NorESM2

The Norwegian Earth System Model version 2 (NorESM2) is developed by Norwegian Climate Center (Seland et al., 2020). Here, we use the "LM" version which has a "low" horizontal resolution (1.9°x2.5°) in the atmosphere. The atmospheric component (CAM6-Nor) is built on CESM2.1 CAM6 but uses a different module (OsloAero6, Kirkevåg et al., 2018) for aerosol chemistry and physics including aerosol-cloud-radiation interactions. CAM6-Nor has 32 vertical layers with a model top around 2.26 hPa. Simulations are performed using fixed monthly SSTs and sea-ice climatology from the year 2000

(ensemble member 1) and from 2010 (ensemble member 2). GHG concentrations represent climatology from the year 2000 with aerosol and aerosol precursor emissions representing year 2019 from CEDS_v2021.

## 2.4 OsloCTM3

OsloCTM3 (Lund et al., 2018;Søvde et al., 2012) is an offline global three-dimensional chemistry transport model driven by 3 hourly meteorological forecast data by the Open Integrated Forecast System (Open IFS, cycle 38 revision 1) at the

European Centre for Medium-Range Weather Forecasts (ECMWF). In this study the model is driven by 2010 meteorology with year 2009 used as spin-up and CEDS_v2021 anthropogenic emissions. The horizontal resolution is ~2.25° × 2.25° and in the vertical 60 layers ranging from the surface up to 0.1 hPa. OsloCTM3 consists of a tropospheric and stratospheric chemistry scheme (Søvde et al., 2012) as well as aerosol modules for sulphate, nitrate, black carbon, primary organic carbon,





secondary organic aerosols, mineral dust and sea salt (Lund et al., 2018). The aerosol radiative forcing is calculated offline

(Myhre et al., 2017) from 3 hourly output from OsloCTM3. For aerosol-cloud interaction only the change in effective radius is simulated, so no rapid adjustment to microphysical properties such as cloud fraction or liquid water content are considered. For the OsloCTM3 we thus report aerosol RF and not aerosol ERF.

## 3 Results

For each model and each ensemble member the ERF due to the 80% reduction in $SO_2$ emission from shipping is calculated

and presented in Fig. 1. The mean values for each individual ensemble range from 0.035 to 0.11 W m$^{-2}$. The ensemble mean ERF for the individual models ranges from 0.057 to 0.089 W m$^{-2}$ (the orange bar in Fig. 1).

As the signal is small, the simulation length needs to be long to assess the ERF in the climate models (Table 1). Also shown in Fig. 1 is the 66%, 90% and 95% confidence interval (CI) calculated from the net radiation at the top of atmosphere over the entire simulation length. For two of the ensemble members the lower end of the 95% CI for the ERFs are less than 0.0.

There is also considerable spread between the individual ensemble members. For two of the models, ModelE_OMA and NorESM2, the mean value for one of the ensemble members is outside of the 66% CI of the other ensemble member.

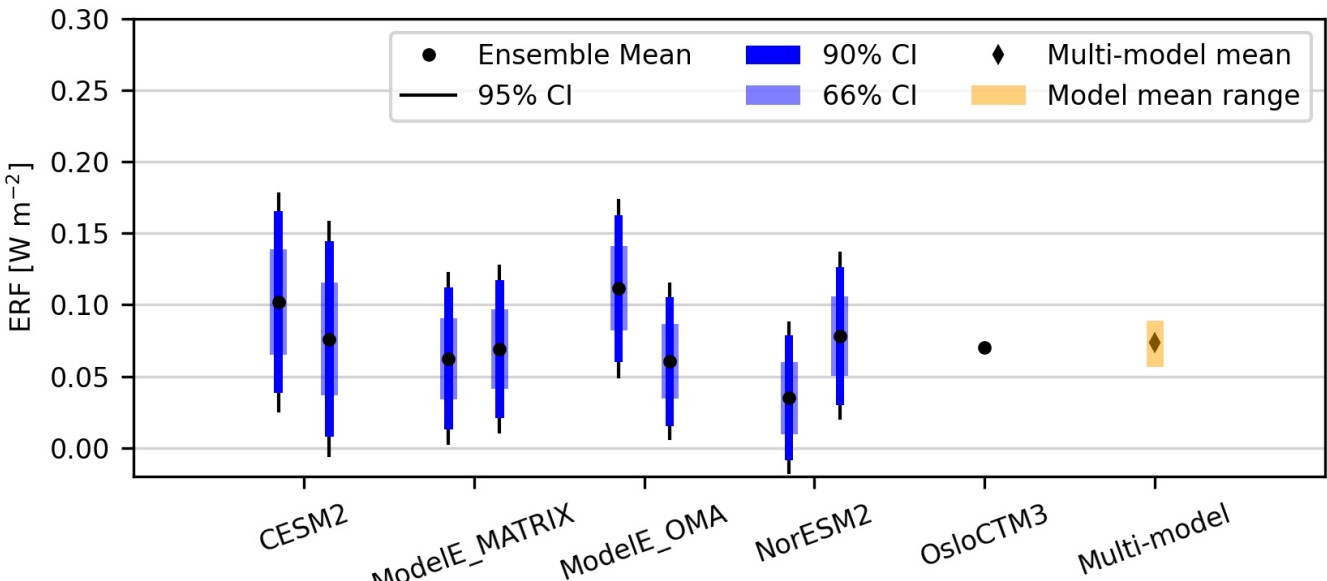

**Figure 1: Effective Radiative Forcing for an 80% reduction in shipping emissions as calculated in the models. Each ensemble**
**member is plotted separately, and the mean ERF value (black dot), the 66% confidence interval (thick coloured bar), 90% confidence interval (thin coloured bar) and 95% confidence interval (vertical solid line) based on the interannual variation are shown. The length of the ensemble members for the individual models are given in Table 1. For OsloCTM3, the calculated RF value is shown as a black dot. The multi-model mean is indicated by a black diamond. The range of the model means (taken as the mean of the ensemble means) is shown as an orange bar.**




For OsloCTM3 the radiative forcing (RF) of the 80% reduction in shipping emissions is shown. The RF is split into RFari of 0.024 W m$^{-2}$ and RFaci of 0.045 W m$^{-2}$, contributing 35 and 65%, respectively, of the total RF of the SO$_2$ emission reduction. The geographical distributions of RFari and RFaci are shown in Fig. S2. Note that the cloud adjustments are not calculated from this model. As shown in Fig. S1, the total anthropogenic emissions of SO$_2$ have decreased over the last three

decades, but with large regional differences (O'Rourke et al., 2021). The simulated one-year change in near-surface mass mixing ratio of sulphate from OsloCTM3 for an 80% reduction in ship emissions has a different geographical distribution than historical sulphur changes as simulated by OsloCTM3 (Skeie et al., 2023) that also show large reduction per year (Fig. S3). Considering longer time periods, the historical change in sulphur overwhelms this one-year drop in shipping emissions in 2020 including over the oceanic regions with the strongest shipping reductions (Fig. S3).

The geographical distributions of the ERF for the individual ensemble members are noisy (Fig. S4) due to internal variability, which is amplified by the weather patterns influencing the source of natural forcers, such as dust and sea salt aerosols. For the model means (Fig. S1, rightmost column) and the multi-model mean (Fig. 2), positive ERF in the North-Atlantic, North-Eastern Pacific and North-Western Pacific are seen. These are areas where the models agree on the sign of ERF, as indicated with dots in Fig. 2.

The strongest ERF is in the North-Atlantic, with values up to 0.78 W m$^{-2}$ in the multi-model mean. Note that the 80% reduction in shipping emissions was applied globally, also in areas with already strict emission control, hence emission reduction in the global ocean may have been even larger than what is used in the simulations and the maximum ERF values underestimated.

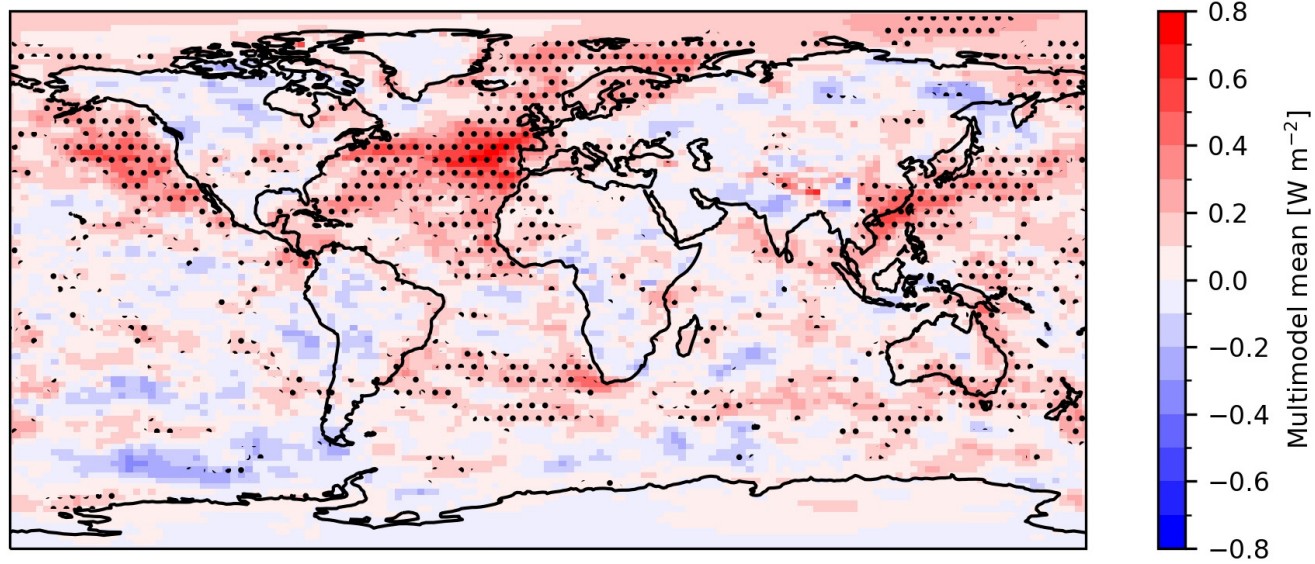

**Figure 2: The multi-model mean of the ERF for 80% reduction in shipping emissions. For each model, an ensemble mean is calculated, then re-gridded to a similar grid before calculating the multi-model mean. Dots indicate areas where at least four out of**



**five models have the same sign of ERF. For OsloCTM3, the calculated RF is included, and note that from this model the RF is always positive.**

## 4. Discussion and conclusion

In this multi-model study, the mean ERF for an 80% reduction in $SO_2$ emissions from shipping from individual models range from 0.057 to 0.089 W m$^{-2}$ with a multi-model mean of 0.073 W m$^{-2}$. Previous estimates of the climate forcing of the IMO shipping regulations in 2020 using single models show a wider range of 0.03 to 0.33 W m$^{-2}$ (Sofiev et al., 2018;Bilsback et al., 2020;Jin et al., 2018;Partanen et al., 2013) .

To calculate the ERF of the 2020 shipping emission regulation in this study, the shipping emission is scaled by a single

factor so that the total emission from the shipping sector is reduced by 80%. This is a simplified test and does not consider that emissions in specific areas already had strict emission regulations prior to 2020. However, there is substantial uncertainty in total sulphur emissions from the shipping sector, for example Eyring et al. (2010) estimated a large uncertainty range of 3 to 10 Tg(S) for the year 2000. Uncertainty in baseline emissions hence causes uncertainty in emission reductions due to the new regulation. Recently, a new CEDS inventory (Hoesly and Smith, 2024) is available that extends emissions

until 2022 (Fig. S1) with a 71% reduction in sulphur emissions from international shipping in 2020 relative to 2019, which is lower than the emission perturbation used in our simulations.  IMO2020 can also be complied with alternate methods, either by switching to low sulphur fuel or by wet scrubbing of the exhaust. Each method results in different physical properties of the exhaust particles and hence produces different impact on clouds (Santos et al., 2024). Such effects are not included in global chemistry models and cannot be derived from emission estimates.

In marine areas, dimethyl sulphide (DMS) is an important sulphur component. Jin et al. (2018) highlighted the significance of the DMS concentration for aerosol-cloud interactions due to shipping emissions. When DMS emissions in their simulations were reduced, the cloud radiative effect of the shipping emission reduction increased. Satellite measurements of DMS from the airborne NASA Atmospheric Tomography (ATom) mission indicate that the DMS emissions in global models may be overestimated (Bian et al., 2024). Therefore, a better representation of the sulphur cycle in marine areas is

needed to further our understanding of the impact of $SO_2$ emission on climate.

The ERFaci due to $SO_2$ emissions from shipping include the formation of ship tracks. The global models are unable to explicitly represent these small scale processes. Watson-Parris et al. (2022) used machine learning to detect all ship tracks in satellite data. They found only a 25% reduction in ship track frequency following the implementation of IMO2020. Shipping emissions also interact with cloud and change cloud properties even if ship tracks are not visible in satellite images

(Glassmeier et al., 2021;Manshausen et al., 2022). Diamond (2023) used a statistical technique to look at the large-scale cloud properties in southeastern Atlantic from satellite images and found a reduction in the magnitude of cloud droplet effective radius and cloud brightening. He estimated the forcing within the shipping corridor from the IMO2020 regulations which implied a global instantaneous radiative forcing due to aerosol–cloud interactions of 0.1 W m$^{-2}$, similar in magnitude as the multi model mean ERF calculated in this study.



Using satellite retrievals, reanalysis wind, ship positions and modelled emissions Manshausen et al. (2023) showed that the cloud droplet numbers respond linearly to ship emissions. On the other hand, the increase in liquid water content is constant over a wide range of emissions perturbations which are caused by compensating effects of increases and decreases in liquid water path in different regimes. They also found that LWP anomalies are largely unchanged before and after 2020, so the liquid water path adjustments were weak due to the ship emission regulation. This indicates that chemistry models without

detailed representation of the liquid water path adjustments do not largely underestimate the ERF of the shipping emission cap. A remaining issue Manshausen et al. (2023) highlighted is how the ship emission regulations have impacted the cloud fraction (Chen et al., 2022;Chen et al., 2024).

Shipping emission regulation has been suggested as a possible reason for the increase in the Earth's energy imbalance as measured by CERES (Clouds and Earth's Radiant Energy System) over the last years, and as a contributor to the acceleration

of global warming (Hansen et al., 2023). The multi-model mean ERF due to IMO2020 is estimated to be 0.073 W m$^{-2}$, with individual model mean ranging from 0.057 to 0.089 W m$^{-2}$. To put the results here in context, an ERF of ~0.1 W m$^{-2}$ for the IMO2020 regulation is comparable to the increase in $CO_2$ ERF of 0.1 W m$^{-2}$ from 2019 to 2022 (Forster et al., 2023). However, it is important to keep in mind that the ERF for the IMO2020 shipping cap calculated in this study has large uncertainties (as for aerosol ERF in general) related specifically to cloud adjustments, emission uncertainties and

uncertainties in the sulphur cycle.

**Code availability**

The code to reproduce the figures in this manuscript: https://zenodo.org/records/11183362 (Skeie, 2024).

**Data availability**

The data needed to reproduce the figures in the manuscript: https://zenodo.org/records/11183362 (Skeie, 2024). The full dataset for the model results will be made available in the NIRD Research Data Archive upon publication. The CEDS emission inventories: https://github.com/JGCRI/CEDS/ (O'Rourke et al., 2021;Hoesly and Smith, 2024).

**Author contribution**

RBS wrote the paper in collaboration with all the co-authors. RBS performed the OsloCTM3 simulations, ØH the CESM2

simulations, RB the NorESM2 simulations, CJ the ModelE simulations. GM did the offline radiative transfer simulations.



**Competing interests**

At least one of the (co-)authors is a member of the editorial board of Atmospheric Chemistry and Physics.

**Acknowledgements**

The work was funded through the Norwegian Research Council project (grant number 314997) and by the European Union's
Horizon 2020 research and innovation program under grant agreement No 820829 (CONSTRAIN project). We would like to thank Dr. Konstantinos Tsigaridis from NASA for his valuable assistance in setting up the NASA GISS ModelE. OsloCTM3, ModelE, CESM2 and NorESM2 simulations were performed on resources provided by Sigma2—the National Infrastructure for High-Performance Computing and Data Storage in Norway (project account NN9188K), and data will be uploaded and shared through their services (project NS9188K) upon publication.

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
