# Peer review of "Multi-model effective radiative forcing of the 2020 sulphur cap for shipping."

_EGUsphere, 2024_

## Author Comment (AC1)

We appreciate the useful comments by the two reviewers. Below follows our responses to the comments by the reviewers and a description of how the manuscript has been modified. The original reviewer's comments are in blue and our response in black.

Anonymous Referee #1

Using global climate model simulations, this study estimates the effective radiative forcing of changes in SO2 due to the IMO 2020 sulfur regulation, which limits the sulfur content in international shipping fuel from 3.5% to 0.5% by mass. The results show the magnitude of the ERF due to 2020 sulfur regulation is 0.073 W m$^{-2}$, which is within the range of 0.03 to 0.33 W m$^{-2}$ in the literature. This study advances our understanding of the ERF magnitude of international shipping induced SO2 changes before and after 2020. Overall, it is a high-quality manuscript with clear representation, robust analysis, and interesting results. I have one major comments for the authors to be addressed:

In the Discussion and conclusion section, the role of DMS in estimation of shipping induced ERF is discussed. I am wondering if the DMS emissions in each model simulations could be derived and compared. The differences of DMS among these model simulations could help explain the spread of the estimated shipping ERF among the ensemble members. I expect to see smaller magnitudes of the ERF in simulations with larger DMS emissions.

We have added an additional table in the manuscript with DMS emissions for the individual model ensembles. The DMS emissions range from 27 to 60 Tg DMS yr$^{-1}$. We have also done radiative transfer calculations on a set of simulations for the IMO regulations using OsloCTM3 model with lower DMS emissions. The RFaci in this set of simulations increased by 23%. In the discussion section, where the role of DMS on the ERF is discussed, these results are incorporated. In the new table, also the burden change (absolute and relative) for the emission perturbations is included. This highlights also other differences in the sulphur cycle in the models than DMS emissions.

The modified paragraph in the discussion and conclusion section:

*"In marine areas, dimethyl sulphide (DMS) is an important sulphur component, and the models included in this study span a large range of natural DMS emissions from 27 to 60 Tg DMS yr$^1$ (Table 2). Jin et al. (2018) highlighted the significance of the natural DMS concentration for aerosol-cloud interactions due to shipping emissions. When DMS emissions in their simulations were reduced, the cloud radiative effect of the shipping emission reduction increased. Satellite measurements of DMS from the airborne NASA Atmospheric Tomography (ATom) mission indicate that the DMS emissions in global models may be overestimated (Bian et al., 2024). A set of simulations with DMS*

*emissions in the OsloCTM3 reduced to 30 Tg DMS $yr^{-1}$, results in similar decrease in sulphate burden (in absolute values) for the 80% emission reduction from the shipping sector while the RFaci increased by 23%. The OsloCTM3 is the model with the largest DMS emissions, but it is also the model with the largest absolute change in sulphate burden among the models included in this study (Table 2). Therefore, a better representation of the sulphur cycle in marine areas is needed to further our understanding of the impact of $SO_2$ emission on climate.»*

Minor comments:

1. Near Line 20: delete "fuel" in "fuel (IMO, 2018)".

Done.

2. Line 20: please cite reference about the 77% reduction. How was this number calculated?

We have replaced the IMO website reference by a reference to the underlying report where this number is calculated from the effect of implementing or not implementing this IMO2020 regulation in 2020.

Corbett, J. J., et al. (2016). Health Impacts Associated with Delay of MARPOL Global Sulphur Standards. https://wwwcdn.imo.org/localresources/en/MediaCentre/HotTopics/Documents/Finland%20study%20on%20health%20benefits.pdf FMI.

3. Line 47: The forcing should be 0.18 W $m^{-2}$ in Jin et al (2018).

In Jin et al they write "Reducing sulfur contents from 3.5 % to 0.5 % could reduce both DRE (from −51.4 to −3.9 mW $m^{-2}$) and CRE (from −0.179 to −0.001 W $m^{-2}$)" 0.18 is only the effect of CRE. If you add the effect of DRE −47.5 mW $m^{-2}$ = −0.0475 W $m^{-2}$ you will get 0.23 W $m^{-2}$.

4. Figure 2: Many dots are broken. I guess the broken dots have the same meaning with the remaining full dots. If so, please correct the broken dots. Otherwise, please explicitly indicate the meaning of the broken dots.

The broken dots were due to the resolution of the grid and the size of the dots. We have replaced the dots by hatching represented by lines and hope this helps.

5. Currently, only two figures in the main text and many figures are organized in the supplementary materials, which makes the audience to refer to the supplementary figures frequently. I suggest the authors move some of the suppl. figures to the main text.

We have moved Figure S2 from the supplement to the main text. In addition, we have combined the ensemble mean for the individual models (rightmost column in Figure S4) with the multi-model mean (Figure 2).

Anonymous Referee #2

This is a multi-model estimate of how the 2020 IMO ship fuel regulation has affected global climate. While the motivation for reduced sulfur fuel content is air quality, the regulation has become an important indicator of human impacts on global climate via aerosols. A constraint on the full impact also helps understand the causes of recent interannual temperature shifts. There have been several evaluations of this ship fuel change but I have seen no multi-model attempts like this and feel the study is fitting for ACP. I do however feel the authors aren't strongly leveraging the benefits of a multi-model approach. I would like to ask the authors if they can provide more assessment of what processes are driving the forcing and its intermodel spread, and for clarification on the quite narrow uncertainty range they estimate.

As outlined in the responses to the specific comments below we have extended our results section and clarified the estimated range, that was not the full uncertainty range but only the spread in the model means.

Major comments

It would be nice if there were deeper context on the model results, as the Results section is quite thin so I feel the study isn't leveraging the breadth of model output the authors have available. For instance I wonder if the authors can provide any information on what processes drive the forcing in the assessed models? E.g. division between ACI and ARI and between RF and ERF. For instance, most models have estimates of cloud radiative effects, which could be used to decompose the contribution of cloud forcing changes, even if 'cloud masking' of non-cloud changes makes this a rough estimate. Second, the GISS model is often run with the Ghan 'clean' (aerosol free) double-radiation call, so if this was done for the simulations here the aerosol direct effect and a more representative cloud radiative forcing can be separated out from the output of what is here treated as two models. Much of the advantage of a multi-model study is that it can be set up to enable process-based comparisons among the models. So I'd encourage the authors to add any indication whether the ERF and its spread are dominated by any particular term, or otherwise to add more of a process-based story to this evaluation.

To add a more process-based story, we have added an additional table to the manuscript for the representation of the sulphur cycle in the model. We find the DMS

emissions in the models to differ (see comments to Reviewer #1) and also the sulphate burden change from the same emission perturbation to be different. This highlight the divergence in the model results in the representation of the sulphate cycle.

In the results section we have added:

*"The 80% reduction in $SO_2$ emissions from the shipping sector is driving these forcing responses. The $SO_2$ emissions chemically react in the atmosphere and convert to sulphate particles that alter the radiation field directly or indirectly via clouds in the atmosphere before the particles are eventually removed through scavenging. Table 2 summarizes the $SO_4$ burden change due to the emission reduction in the shipping sector from the individual ensemble members in each model. The $SO_4$ burden change ranges from -0.018 to -0.07 Tg (-1.5 to -2.7%) for the same emission perturbation of 8.7 Tg $SO_2$ $yr^1$ indicating that differences in the representation of the sulphur cycle contribute to the model spread in the forcing."*

It is challenging to decompose the ERF calculated in models, and the GISS model was unfortunately not run with the double-radiation call. As the GISS models and OsloCTM3 do not include the cloud adjustments, we have highlighted this in the discussion and results sections and presented results of RF and ERF separately. The results from the models calculating RF and those calculating ERF are similar.

In the results section we have added:

*"For OsloCTM3, ModelE_MATRIX and ModelE_OMA the cloud adjustments to the shipping emission perturbations are not included (see Method) and what we report for these models is the RF. Their mean RF ranges from 0.066 to 0.086 W $m^{-2}$, similar to the range for NorESM2 and CESM2 of 0.057 to 0.089 W $m^{-2}$, which includes the cloud adjustments. This might indicate that cloud adjustments play a limited role in this forcing in CESM2 and NorESM2. "*

And in the discussion and conclusion section:

*"The models included in this study have a variable degree of microphysical cloud adjustments. OsloCTM3, ModelE_GISS and ModelE_MATRIX do not include changes in liquid water path and cloud cover and hence RF is reported. The models reporting RF and the models reporting ERF show a similar spread, indicating a limited role of cloud adjustments in the two other models."*

The stated ERF range of 0.06-0.09 Wm-2 is quite narrow and at odds with the "large uncertainties" assertion in the manuscript's last sentence. So I wonder if this is an accurate portrayal of the uncertainty. For one, each of the models in Fig. 1 has a larger uncertainty than the multi-model uncertainty, which gives the sense there is more uncertainty than depicted here. I wonder if the authors can instead give an uncertainty range that combines the evaluated factors?

Regarding the narrow range of the multi-model results, the range presented in the abstract is the model mean range. Clearly the uncertainty is larger than what the spread in the model means represents. However, it is difficult to assess the full range of the uncertainty in the ERF due to the IMO regulation of 2020 (including uncertainty in emissions, sulphur cycle, forcing and cloud adjustments). Therefore, we do not attempt to come up with an uncertainty range on our estimates but we have added a sentence to the abstract highlighting the large uncertainties discussed in the paper.

*"The full uncertainty in the ERF due to the new regulation is not quantified but will very likely be high considering the contribution of uncertainties in shipping $SO_2$ emissions, the sulphur cycle, the modelling of cloud adjustments and the impact of interannual variability on the method of calculating radiative forcing."*

In the abstract we have also tried to make this clearer by adding the word individual:

*"The individual model means range from 0.06 to 0.09 W $m^{-2}$ corresponding to the ERF due to the increase in $CO_2$..."*

We have also added a clarification at the end of the first paragraph in the discussion section:

*"The uncertainty calculated based on the interannual variability in the individual ensemble members is larger than the spread in the model mean estimates (Fig. 1). This highlights the importance of simulations with sufficient length for the ERF calculations. In addition to uncertainties related to interannual variability in the simulations, there are additional uncertainties as discussed below."*

Also, I wonder if the authors can comment on features of the models that make their ERFs similar. For instance, if the ACI RF (first indirect effect) were the leading factor, could it be that the models have similar CCN parameterizations? The Abdul-Razzak and Ghan parameterization is ubiquitous, which might explain some of the intermodel similarity.

In the model description, we have added descriptions of the CCN parameterization in the models. CESM2, ModelE_MATRIX and NorESM2 are all based on the Abdul-Razzak and Ghan parameterization.

CESM2: "*cloud droplet activation parameterization is based on Abdul-Razzak and Ghan (2000)*»

ModelE: "*In MATRIX, the cloud droplet activation parameterization is based on Abdul-Razzak and Ghan (2000), while in OMA, the aerosol conversion into cloud condensation nuclei is empirical, following the method outlined by Menon and Rotstayn (2006).*"

NorESM2: *"As in both CESM2 and MATRIX, the cloud droplet activation parameterisation in CAM6-Nor is based on Abdul-Razzak and Ghan (2000) (see also Kirkevåg et al., 2013)."*

In the discussion section we have added the following sentence:

"Note also that there are similarities in the cloud droplet activation parameterization in the models (see Method), that may reduce the model spread."

Could the authors please add context on how their results are different or improved from other recent studies on the climatic impact of the ship fuel regulation?

In the first paragraph in the discussion and conclusion section, we have highlighted that previous single model studies have used different assumptions of the emission reductions due to shipping, while in this study we use multiple models with similar set-up of the perturbation. We have also highlighted the importance of long enough simulations for quantifying the forcing.

Added text: *"The uncertainty calculated based on the interannual variability in the individual ensemble members are larger than the spread in the model mean estimates (Fig. 1). This highlights the importance of simulations with sufficient length for the ERF calculations. In addition to uncertainties related to interannual variability in the simulations, there are additional uncertainties as discussed below"*

Currently much of the Discussion section summarizes past studies, and feels more suited to the Introduction. I'd like to see more context on how the current study advances the field.

We have added more content based on the results of this paper to the discussion section. Both in the paragraph regarding DMS emissions (see response to Reviewer #1) and ERF vs. RF as described above.

Specific comments

Lines 25-6: I'd like the ERF and RF (defined in Line 140 but first used in 72) to be explained in the Introduction, especially since "effective radiative forcing" is prominently in the title. Can the authors please briefly describe the terms in each, and which are expected to be relevant? For instance, is it expected for the semi-direct effect (the ERF ARI – RF ARI) to be a contributor, or is sulfate not sufficiently absorbing?

Following the definition of ERF, ERFari and ERFaci, we have added the following sections defining RF and describing the different terms:

*"The ERF metric includes the radiative effects of atmospheric adjustments to the initial forcing that are not mediated by surface temperature change. Such so-called 'rapid adjustments' include changes in stratospheric temperature, tropospheric temperature, water vapour, surface albedo and clouds (Smith et al., 2018;Boucher et al., 2013;Sherwood et al., 2015). Changes in cloud liquid water path and cloud cover comprise the main cloud adjustments, and are associated with large uncertainties (Bellouin et al., 2020). When all tropospheric adjustments are excluded, (i.e., only stratospheric temperatures adjust), the forcing is termed radiative forcing (RF). This can be divided into RF of aerosol-radiation-interaction (RFari), which represents the interaction of aerosol particles themselves with the radiation field, and RF of aerosol-cloud-interaction (RFaci), which accounts for how aerosol particles change the reflectivity of clouds by altering the cloud droplet number concentration. The adjustments following aerosol-radiation-interaction for a sulphate perturbation are negligible as sulphate aerosols are non-absorbing (Stjern et al., 2023)."*

The semi-direct effect is not expected to contribute as sulphate is not absorbing. In the next section, based on IPCC AR6 we have stated that ERFaci is the dominant contributor to aerosol ERF due to anthropogenic SO2 emissions. However, in the added section (see above), we have highlighted the large uncertainties in the adjustment terms.

Line 29: The cited "total aerosol ERF" as stated in the reference refers to the change "over the industrial era (1750–2014)". Since this doesn't significantly include natural aerosol, the description would be clearer by specifying this as a "total anthropogenic aerosol ERF" or similar. Natural aerosols can also be assessed to have an ERF.

We have added "*over the industrial era (1750–2014)*" to the sentence to clarify. The total aerosol ERF includes for instant forcing due to biomass burning emissions, that can both be of natural and anthropogenic origin.

Line 34: "Assuming" makes it sound like the 80% reduction was a completely arbitrary choice. Maybe "approximating" is better, or this can be reworded another way?

Rewritten as: "Approximating the reduction of $SO_2$ emissions in 2020 from the shipping sector to 80%,"

Lines 43-51: Because there is already a range of forcing estimates for the same case, can the authors please briefly say in the Introduction how the present study is an improvement or at least a worthwhile addition to this literature?

Previous studies have used different setups to calculate the forcing estimates, different baseline emissions, different size of emission reduction. Here we use the same setup for all models which is an advantage. We have added in the first paragraph of the Discussion and conclusion section that previous studies have used different

assumptions regarding emissions reductions, which is also highlighted in the introduction. In the last sentence in the Introduction we have added that the emission perturbation is from the same baseline.

Methods, generally: Can the authors please add a bit of info on whether the aerosol radiative effects and/or CCN activation schemes are different between the models in any way that matters? Conversely, if these models are highly similar, the intermodel range might not represent an accurate representation of current process uncertainties.

We have added additional information on the CCN activation schemes in the model section (see response above).

We have also (as mentioned above) clarified that the intermodel range do not represent current process uncertainties. The uncertainty is larger.

Line 63: When the authors say they "perform two atmosphere-only simulations", for most models they seem to actually mean four, given the two ensemble members. Maybe change to "two types of atmosphere-only" simulations or clarify this another way?

Added suggested change.

Line 72: RF is used here but has not yet been defined.

Definition moved to the introduction.

Lines 72-4: Can the authors please briefly explain in the text why they have chosen to use a CTM? Is there an added benefit over the GCMs, which presumably could be nudged in a way that mimics the CTM's being driven by meteorology? Or it's predominantly just to have one more model?

Using a CTM avoids noise from internal variability and the RF can be calculated based on a single year of simulation, as the meteorology is exactly the same in the control and perturbed run. In a multi model study it is also an advantage to have several models.

In the model description we have added a sentence regarding the CTM: "*Results from a CTM are not influenced by noise and therefore simulation length of one year is sufficient.*"

Table 1 caption: Can the authors please say in the manuscript why the CTM is only one year? Is this because the CTM is constrained by meteorological inputs and hence less susceptible to noise? I think this would look better in the text rather than the caption, but leave it to the authors to decide.

We keep the text in the Table 1 caption but added a sentence to the Method section (see response above).

Lines 85-6: The same info on the climatology seems to be repeated for each model, as it appears in Lines 99-100 and Lines 109-110. I'd encourage the authors to avoid repeats by describing common setup information at the start of the Methods rather than in each model description.

We have moved these to a common description at the start of the Method section.

*"We run two ensemble members for each global climate model, which differ in terms of the climatology of sea-surface temperature and sea-ice concentrations that are used in the simulations (ensemble member 1 uses 2000 climatology and ensemble member 2 uses 2010 climatology)."*

Lines 92-7: The ModelE citations are confusing. Since the version used is E2.1, can the authors please cite Bauer et al 2020 more centrally and omit references to older model versions not used here, which are cited in the Bauer paper anyhow?

We have removed the old references to Bauer et al 2008, Koch et al 2006 and Nazarenko et al., 2017, and refered to Bauer et al 2020 as it is sufficient.

Line 98: Can the authors please explain why/how they expect the two ModelE versions to perform differently? Does this mostly stem from differences in sulfate size from its fixed value in OMA (which ideally would be stated) and the interactive value in MATRIX?

Yes, with a predicted aerosol size distribution (MATRIX), compared to prescribed constant aerosol size distribution (OMA), aerosol-cloud-radiation interactions are different.

Line 97: Do the authors expect the lifetime effect (which should be briefly described) would contribute much? There's a brief reference near the end of the manuscript but I feel this should be indicated sooner.

In the new section in the introduction, we have introduced the cloud adjustment terms (see response above).

In the results section we have added a new paragraph where we compare the results with the models including the cloud adjustments and the models without (see response to Major comment above). In the discussion section, we have added more content to the section on the cloud liquid path related to our results (see response to Major comment above), and included a new sentence referring to another study at the end:

*"The recent study by Yuan et al. (2024) found the cloud fraction adjustments to contribute by 60% to their forcing estimate of 0.2 W m$^{-2}$ (for the global ocean) combining satellite data and global modelling, while liquid water path adjustments were negligible on the global scale. "*

The liquid water path adjustment seems to play a smaller role from comparing the models including and excluding these adjustments. Also, the studies mentioned in this

paragraph indicate a smaller role of the liquid water path adjustments, however there cloud fraction adjustments are uncertain.

Line 118: Is the OsloCTM3 aerosol module one-moment or two?

It is a one-moment aerosol module for sulphate.

Fig. 1: Does it really make sense for the range from multiple models to be smaller than the range from any single model? I would expect there to be compounding uncertainties.

As indicated in the figure legend, what is shown is the model mean range, which is not the uncertainty range. As stated above, we have added an additional sentence regarding uncertainties to the abstract and the first section of the discussion and conclusion. Hope this is clearer now.

Lines 142: Can the authors estimate RF ARI and RF ACI in any other models for comparison to these OsloCTM3 values? Or generally make any apples-to-apples comparisons between the models for anything other than the full simulated effect?

Regarding radiation, it is difficult to properly compare any other effect than the full simulated effect. But we have added results on changes in the sulphur cycle, where the results show different sulphate perturbations for the same emission perturbation.

Fig. 2: Given there are only two figures in the manuscript, it would make sense to at least make this a compound figure with one of the Supplementary figures, if there isn't anything more directly relevant to show here.

We have merged the model mean (rightmost column in Fig S4) with Fig 2. We have also moved Fig S2 from the supplement to the main text. In addition, we have added a table to the main text regarding the changes in sulphate.

Line 168: Please give a brief rationale of why this study, which generally agrees with the others, is different or original from the previous attempts.

We have indicated that previous studies assume different emission reductions, as also highlighted in the introduction. Here we use multiple models with the same emission background and the same emission perturbation. We also highlight here the importance of long simulations, and uncertainties related to internal variability in the simulations.

Lines 180-185: I find this paragraph confusing to read. This makes it sound like DMS is being released from ships, but these references are about naturally emitted DMS, right? I'd like to see this more clearly delineated.

Added «natural" to make this clearer.

Line 195: "modelled emissions" of what?

"modelled shipping emissions"

Lines 195-6: "showed that the cloud droplet numbers respond linearly" to "showed that cloud droplet number responds linearly"

We have removed this sentence to focus more on the adjustment in this paragraph.

Line 200: I think the "liquid water path adjustments" is the second indirect effect that is indirectly hinted at when Line 97 mentions the GISS model has only a "first indirect effect" (Line 97), but I'd like to see this effect briefly explained early in the article and then described consistently.

In the introduction, the adjustments are now clearly defined (see response above). We have also more clearly stated in the method and results sections that the GISS models and OsloCTM3 do not include the liquid water path adjustments and thus calculate RF and not ERF (see response above). We have tried to be more consistent and avoid the term first indirect effect. We have rewritten in the ModelE description:

 *"MATRIX and OMA only include the effect of aerosol on the cloud droplet concentrations  (Bauer et al., 2020), and hence not changes in the cloud liquid water path."*

At the end of the method section, we have also added: *"Although cloud adjustments are not included in all models, we present the results collectively as ERF."*

Line 202: Is the cloud fraction impact separate or heavily linked to the liquid water path adjustment? Clouds with less cloud fraction tend to have less liquid water path if not normalizing by the fraction. I see this is following the language of observational studies, but find it a bit confusing.

These processes are not fully understood. We have added "in observational based studies" here.

Line 208-9: The stated "large uncertainties" are what I'd expect but this article's main conclusion is a 0.06-0.09 Wm-2 uncertainty range, which is quite small. Please reconcile.

We do not state that 0.06-0.09 is the uncertainty range. It is the range of the model means, and clearly more narrow than the full uncertainty range. We have added a sentence to the abstract and first part of the Discussion and conclusion section clarifying this (see response above).

Typographic or minor errors

Line 63: "is used to" to "are used to"

Done.

Line 186: "include" to "includes"

Done.

Line 195: "reanalysis wind" to "reanalysis winds"

Done.

Table 1: "#Ensemble members" could at least be "# of ensemble members".

Done.

**References**

Abdul-Razzak, H., and Ghan, S. J.: A parameterization of aerosol activation: 2. Multiple aerosol types, J. Geophys. Res., 105,6837-6844, https://doi.org/10.1029/1999JD901161, 2000.

Bauer, S. E., Tsigaridis, K., Faluvegi, G., Kelley, M., Lo, K. K., Miller, R. L., Nazarenko, L., Schmidt, G. A., and Wu, J.: Historical (1850–2014) Aerosol Evolution and Role on Climate Forcing Using the GISS ModelE2.1 Contribution to CMIP6, J. Adv. Model. Earth Syst., 12,e2019MS001978, https://doi.org/10.1029/2019MS001978, 2020.

Bellouin, N., Quaas, J., Gryspeerdt, E., Kinne, S., Stier, P., Watson-Parris, D., Boucher, O., Carslaw, K. S., Christensen, M., Daniau, A. L., Dufresne, J. L., Feingold, G., Fiedler, S., Forster, P., Gettelman, A., Haywood, J. M., Lohmann, U., Malavelle, F., Mauritsen, T., McCoy, D. T., Myhre, G., Mülmenstädt, J., Neubauer, D., Possner, A., Rugenstein, M., Sato, Y., Schulz, M., Schwartz, S. E., Sourdeval, O., Storelvmo, T., Toll, V., Winker, D., and Stevens, B.: Bounding Global Aerosol Radiative Forcing of Climate Change, Rev. Geophys., 58,e2019RG000660, 10.1029/2019RG000660, 2020.

Bian, H., Chin, M., Colarco, P. R., Apel, E. C., Blake, D. R., Froyd, K., Hornbrook, R. S., Jimenez, J., Jost, P. C., Lawler, M., Liu, M., Lund, M. T., Matsui, H., Nault, B. A., Penner, J. E., Rollins, A. W., Schill, G., Skeie, R. B., Wang, H., Xu, L., Zhang, K., and Zhu, J.: Observationally constrained analysis of sulfur cycle in the marine atmosphere with NASA ATom measurements and AeroCom model simulations, Atmos. Chem. Phys., 24,1717-1741, 10.5194/acp-24-1717-2024, 2024.

Boucher, O., D. Randall, P. Artaxo, C. Bretherton, G. Feingold, P. Forster, V.-M. Kerminen, Y. Kondo, H. Liao, U. Lohmann, P. Rasch, S. K. Satheesh, S. Sherwood, B. Stevens, and Zhang, X. Y.: Clouds and Aerosols, in: Climate Change 2013: The Physical Science Basis. Contribution of Working Group I to the Fifth Assessment Report of the Intergovernmental Panel on Climate Change edited by: Stocker, T. F., D. Qin, G.-K.

Plattner, M. Tignor, S. K. Allen, J. Boschung, A. Nauels, Y. Xia, V. Bex, and Midgley, P. M., Cambridge University Press, Cambridge, United Kingdom and New York, NY, USA, 2013.

Jin, Q., Grandey, B. S., Rothenberg, D., Avramov, A., and Wang, C.: Impacts on cloud radiative effects induced by coexisting aerosols converted from international shipping and maritime DMS emissions, Atmos. Chem. Phys., 18,16793-16808, 10.5194/acp-18-16793-2018, 2018.

Kirkevåg, A., Iversen, T., Seland, Ø., Hoose, C., Kristjánsson, J. E., Struthers, H., Ekman, A. M. L., Ghan, S., Griesfeller, J., Nilsson, E. D., and Schulz, M.: Aerosol–climate interactions in the Norwegian Earth System Model – NorESM1-M, Geosci. Model Dev., 6,207-244, 10.5194/gmd-6-207-2013, 2013.

Menon, S., and Rotstayn, L.: The radiative influence of aerosol effects on liquid-phase cumulus and stratiform clouds based on sensitivity studies with two climate models, Clim. Dyn., 27,345-356, 10.1007/s00382-006-0139-3, 2006.

Sherwood, S. C., Bony, S., Boucher, O., Bretherton, C., Forster, P. M., Gregory, J. M., and Stevens, B.: Adjustments in the Forcing-Feedback Framework for Understanding Climate Change, B. Am. Meteorol. Soc., 96,217-228, 10.1175/BAMS-D-13-00167.1, 2015.

Smith, C. J., Kramer, R. J., Myhre, G., Forster, P. M., Soden, B. J., Andrews, T., Boucher, O., Faluvegi, G., Fläschner, D., Hodnebrog, Ø., Kasoar, M., Kharin, V., Kirkevåg, A., Lamarque, J. F., Mülmenstädt, J., Olivié, D., Richardson, T., Samset, B. H., Shindell, D., Stier, P., Takemura, T., Voulgarakis, A., and Watson-Parris, D.: Understanding Rapid Adjustments to Diverse Forcing Agents, Geophys. Res. Lett., 45,12,023-012,031, 10.1029/2018GL079826, 2018.

Stjern, C. W., Forster, P. M., Jia, H., Jouan, C., Kasoar, M. R., Myhre, G., Olivié, D., Quaas, J., Samset, B. H., Sand, M., Takemura, T., Voulgarakis, A., and Wells, C. D.: The Time Scales of Climate Responses to Carbon Dioxide and Aerosols, J. Clim., 36,3537-3551, https://doi.org/10.1175/JCLI-D-22-0513.1, 2023.

Yuan, T., Song, H., Oreopoulos, L., Wood, R., Bian, H., Breen, K., Chin, M., Yu, H., Barahona, D., Meyer, K., and Platnick, S.: Abrupt reduction in shipping emission as an inadvertent geoengineering termination shock produces substantial radiative warming, Communications Earth & Environment, 5,281, 10.1038/s43247-024-01442-3, 2024.